# Management of Double Sensitization to Vespids in Europe

**DOI:** 10.3390/toxins14020126

**Published:** 2022-02-08

**Authors:** Berta Ruiz-Leon, Pilar Serrano, Carmen Vidal, Carmen Moreno-Aguilar

**Affiliations:** 1Allergy Section of University Hospital Reina Sofia-IMIBIC, ARADyAL Network, National Institute of Health Carlos III, 14005 Cordoba, Spain; mb.ruiz.sspa@juntadeandalucia.es (B.R.-L.); carmen.moreno.sspa@juntadeandalucia.es (C.M.-A.); 2Allergy Department of Complejo Hospitalario Universitario de Santiago de Compostela, 15706 Santiago de Compostela, Spain; Carmen.Vidal.Pan@sergas.es

**Keywords:** *Vespula*, *Polistes*, allergens, double sensitization to vespids, cross-reactivity, venom immunotherapy

## Abstract

Wasp allergy with a diagnostic profile of double sensitizations to vespid venom is a frequent clinical problem in areas where different genera of wasps are present. Identification of the insect responsible for serious reactions poses a diagnostic challenge as the only effective treatment to date is immunotherapy based on the specific venom. In southern Europe, the double sensitization to *Vespula* and *Polistes* venoms is highly frequent. It has been shown that the major allergenic proteins (Phospholipase A1 and Antigen 5) share sequences across the different genera and species, which would be the cause of cross-reactivity. Additionally, the minor allergens (Dipeptidyl-peptidases, Vitellogenins) have been found to share partial sequence identity. Furthermore, venom contains other homologous proteins whose allergenic nature still remains to be clarified. The traditional diagnostic tools available are insufficient to discriminate between allergy to *Vespula* and *Polistes* in a high number of cases. IgE inhibition is the technique that best identifies the cross-reactivity. When a double sensitization has indeed been shown to exist or great uncertainty surrounds the primary sensitization, therapy with two venoms is advisable to guarantee the safety of the patient. In this case, a strategy involving alternate administration that combines effectiveness with efficiency is possible.

## 1. Introduction

Humans develop specific IgE antibodies to wasp venom proteins after being stung by them. Thus, depending on the wasp species living in an environment, allergic sensitization could change [1]

The *Vespidae* of the hymenoptera are represented by six sub-families, four of which are present in Europe: *Masarinae*, *Eumeniae*, *Polistinae* and *Vespinae*, although the first two follow a solitary behavior, and thus do not pose allergy problems. The females of *Polistinae* and *Vespinae* are anatomically equipped for stinging and the injection of venom. The taxonomic features of these insects include similarities and differences between genera and species, both on the morphological and functional levels. The allergenic composition of the venom also depends on the taxonomy, although to date this aspect has only been partially studied [2] the allergology literature, it is commonly claimed that *Vespula* is the wasp of northern Europe, and *Polistes* that of southern Europe, although this statement requires some qualification. The *Vespinae* sub-family includes the genera *Vespa*, *Vespula* and *Dolichovespula* (Figure 1).

*Vespula* is widely distributed across Europe, in particular *Vespula germanica*, which is present across the whole continent and the archipelagos, including the Canary Islands and the Azores, due to its potentially highly invasive behavior and its ability to adapt to a wide range of habitats and climates [3]. *Vespa* generally occupy habitats restricted to wooded and mountainous areas, but *Vespa velutina*, originating from eastern Asia, has widely colonized several European countries [4] and has given rise to health problems even in urban areas [5] *Dolichovespula* is widely distributed in Europe, but with scant presence, limited to mountain areas. *Polistes* is particularly abundant in the temperate areas of southern Europe and the Mediterranean, although some species of *Polistes* such as *P*. *biglumis*, *P nimpha* or *P atrimandibularis* are found in Russia, Poland, Scandinavia and the Baltic Republics [6]. These data suggest that in any area in Europe, wasps may coexist, at least from the genera *Vespula* and *Polistes* although there may be significant differences in the number of individuals at each location. In Europe, to date, a considerable sensitization to the venom of *Vespula* and *Polistes* has been observed in the Mediterranean regions while in the north and center of Europe sensitization to *Polistes* has been considered infrequent. Nevertheless, in central Europe the allergy to *Polistes* could be underdiagnosed, as it does not form part of normal clinical practice.

In the south of Europe, the double sensitization to *Vespula-Polistes* is frequent [7,8,9], more frequent even than the double sensitization to *Apis-Vespula* [10]. In Spain, the double positivity to *Vespula-Polistes* has been found to range from 50.5% to 61.5% in the multiple studies conducted [11,12,13,14]. The fact that double sensitizations are frequent represents a clinical and diagnostic problem when facing allergic patients.

Figure 2 shows the three allergenic vespids with the most impact in recent years in Europe.

The aim of the present review is to analyze available data on this clinical problem in order to help physicians to better understand and decide specific treatment for their patients.

A narrative revision was made. The search was based on PUBMED, EMBASE and The COCHRANE Library; in a complementary way, www.allergen.org (accessed on 12 December 2021), www.allergome.org (accessed on 12 December 2021) and www.seaic.org (accessed on 9 December 2021) were consulted to obtain specific information on allergens and insects. Results were filtered by English language and date of publication 2000 and ahead. The terms introduced in the advanced search were: allergy to vespid/hymenoptera venoms; double sensitization to vespids; venom allergens/allergenic proteins; cross-reactivity hymenoptera and venom immunotherapy. Any additional information provided by authors was also considered.

## 2. Venoms from Different Species Share Similar Allergens: Cross-Reactivity

To identify allergic patients, serum specific IgE (sIgE) determinations should be performed. Allergic individuals recognize sIgE against some proteins in the venom. The sIgE recognition allows doctors to identify the specific insect involved. However, the problem arises when sIgE determinations are no specific or due to similar proteins in different venoms which is known as cross-reactivity [8,11,15,16]. The main problem of cross-reactivity is the undetermined identification of the culprit insect, leading to misdiagnosis and therefore mistreatment [17,18]. An indeep knowledge of venom composition is mandatory to understand the sensitization profile of patients.

Among the venoms from the insects of the *Vespidae* family those best characterized from the allergological point of view are those from *Vespula vulgaris* and *Polistes dominula* with five allergens having been recorded to date in each [19,20]. In recent years, with the expansion of *Vespa velutina* in Europe, the allergenic capacity of its Vesp v 5 and Vesp v 1 allergens has been established in patients experiencing anaphylaxis after a sting by this hymenoptera [21,22]. Table 1 shows the main data on the venom from *Vespula* and *Polistes*.

As it can be seen in Table 1, some allergens are glycosylated. The presence of carbohydrates (CBHD) could explain cross-reactivity together with the partial identity between the protein fractions, either in the sequence or in the structures resulting from folding.

In the case of vespids, the cross-reactivity between *Vespinae* and *Polistinae* may be explained by the recognition of homologous proteins present in both venoms [24] (Table 2), as the venoms from the *Polistes* genus lack glycosylated residues [25]. Noteworthy are the main allergens such as Phospholipase A1, which have been hypothesized to play a role in the recognition of different isoforms [26], or Antigen 5 with the existence of a pronounced cross-reactivity between families [27]. Additionally, the minor allergens such as the hyaluronidases, the dipeptidyl peptidases or the vitellogenins may be the cause of cross-reactivity [25,26].

The allergenic composition of the venoms from the *Vespinae* sub-family is very similar and a high degree of cross-reactivity exists between the different genera (*Vespula*, *Vespa*, *Dolichovespula*) [28]. The clinical relevance of this cross-reactivity is reflected in fact that a patient primarily sensitized to *Vespula* may experience a severe reaction after being stung by *Vespa/Dolichovespula* and vice versa, and further still, a patient allergic to the latter can be adequately treated with *Vespula* venom [27]. A high degree of similarity exists in the allergens from the different species of the *Polistes* genus in Europe, although important allergenic differences between the European (*Polistes dominula*, *Polistes gallicus*) and American species (*Polistes annularis*, *Polistes exclamans*, *Polistes fuscatus*, *Polistes metricus*) have been reported, as they belong to different subgenera [29,30]. This has meant that in Europe, an extract from de *Polistes dominula* has been used for diagnosis and treatment, while in the United States, an extract from *Polistes* spp., a mixture from five regional species, is normally used.

### 2.1. Phospholipase A1 (Ves v 1; Pol d 1)

These important allergens in vespid venom hydrolyze the SN-1 position of phospholipids and show hemolytic activity [31]. Their biological action is important for the allergenic characterization of venom extracts, since allergenic vaccines are made from the source of natural venoms and identification of individual proteins based on its biological activity is a key point to standardize the content batch to batch.

Ves v 1 is a non-glycosylated allergen of 35 kDa. It has been produced recombinantly in eukaryotic media [29]. Pol d 1 (isoforms Pol d 1.0101, Pol d 1.0102, Pol d 1.0103, Pol d 1.0104) has a mass of 34 kDa and is the most frequent allergen in patients sensitized to *Polistes dominula* [26].

Phospholipases A1 in their natural form present sequence homology between each other, and partial identities to a varying degree have been reported (these are normally greater in insects which are taxonomically more closely related) [24], as can be seen in Table 3.

Recently, the similarity of phospholipase A1 from *Vespula vulgaris* and *Polistes dominula* has been reported to be 76% and 71%, respectively [32]. Unlike what occurs with other phospholipases A1, Vesp v 1 is a glycosylated allergen [33]. This is relevant due to the possible interference of an sIgE against CBHD in the diagnosis, as occurs with other hymenoptera allergens. However, as there is no commercially available diagnostic tool to identify the specific IgE against this allergen, as the problem does not actually exist in clinical practice.

### 2.2. Hyaluronidase (Ves v 2; Pol d 2)

This is a common component of hymenoptera venom. It hydrolyzes hyaluronic acid from the extracellular matrix, thus favoring the propagation of the venom beyond the sting site. It is capable of initiating a pathogenic reaction in most allergic patients [34].

The hyaluronidases are glycosylated allergens that show cross-reactivity between each other, apart from that caused by the presence of CBHD, due to partial protein sequence identity. This also affects Apoidea. Partial sequence identity has been reported for the natural forms of Ves v2 (isoforms Ves v 2.0101, Ves v2.0201), Ves g 2, Pol a 2, Dol m 2, Api m2 and Api c 2 [20], rising up to 90–95% among species from *Polistes* and *Vespula* [19] Ves v 2.0201 (inactive) seems to be the dominant isoform in this venom [29]. It has been suggested that its N-glycan structures are necessary for the recognition of IgE and IgG antibodies, but are not required to produce T-cell reactivity in vitro [35], which could be of greater importance to producing the allergic disease than to contributing to the effect of the immunotherapy.

### 2.3. Dipeptidyl Peptidases IV (Ves v 3; Pol d 3)

These are aminopeptidases that cleave dipeptides from the N-terminal ends of polypeptides that contain proline or alanine in the penultimate position [36], thus activating or inactivating the substrate.

Ves v 3 and Pol d 3 are major glycosylated allergens of 100 kDa that have been produced recombinantly. Both proteins have a 76.1% sequence identity, which explains their high cross-reactivity. Homology with the dipeptidyl peptidase of bee venom has also been demonstrated [37], which could offer a diagnostic opportunity, as Pol d 3 is not commercially available for the determination of sIgE, although it is for rApi m 5.

### 2.4. Antigen 5 (Ves v 5; Pol d 5)

These are the most important allergens and the most abundant proteins in wasp venom. They belong to the CAP superfamily (cysteine-rich secretory proteins, Antigen 5 and proteins related to pathogenesis-related 1), but their function in hymenoptera venom still remains unknown.

Ves v 5 and Pol d 5 are non-glycosylated proteins of 23 kDa, which have been produced recombinantly. Ves v 5 in its natural form shows sequence homology with Ves g 5, Ves m 5, Ves f 5, Ves p 5, Ves s 5, Ves vi 5, Pol a 5, Pol d 5, Pol e 5, Pol f 5, Pol g 5, Dol a 5, Dol m 5, Vesp c 5, Vesp m 5, Pac c 3, Poly s 5, Sol i 3 and Sol r 3 [20]. Recently, it has been established that there is a 77% and 75% similarity between the antigen 5 from Vespa velutina venom and antigen 5 from the common wasp (*Vespula vulgaris*) and the European paper wasp (*Polistes dominula*) [32].

### 2.5. Vitellogenin (Ves v 6)

These are glycosylated proteins of 200 kDa, which appear to play an important role as IgE sensitizers, although their natural function is not known. Ves v 6 shows a 60.15% sequence identity with the equivalent protein from *Polistes dominula*, although it has not been reported as an allergen [38].

### 2.6. Serine Protease (Pol d 4)

Its role in *Polistes dominula* is probably related to its ability to initiate coagulation and injure tissue. It has been described as an allergen that is capable of sensitizing only a minority of patients. Although a homologous allergen has not been reported in *Vespula*, the corresponding protein has been 93% sequenced both in *Vespula germanica* and in *Vespula vulgaris*, with a reported sequence identity of 51.9% and 52.6%, respectively. The allergenic capacity of serine proteases from Apoidea venom has also not been reported, but they show a 35–40% sequence identity with Pod d 4 [24,39].

### 2.7. Other Components

Recently, the venom from these vespids has been analyzed from the proteomic point of view with the aim of making diagnostic and therapeutic advances. Forty-seven proteins with a known function were identified in the venom of *Vespula* and 25 in the venom from *Polistes*, with 14 proteins overlapping in both venoms. All allergens reported to date were included with the exception of Pol d 4. New proteins such as Phospholipase A2 were identified in both venoms while others, such as Mastoparan, Vespakinin or Vespulakinin, were only found in *Vespula* venom. Icarapin, lethal essential for life-like protein (LELLP) or vascular endothelial growth factor C were exclusively recognized in *Polistes*. No great differences in the molecular function and biological processes of the proteins were observed between both venoms [31].

## 3. Tools to Identify the Culprit Insect: Helping with the Diagnosis

The main difficulty in the clinical management of patients doubly sensitized to vespid venom is the diagnosis when the presence of two or more allergenic vespids has been detected in the area. The characteristic profile is that of a patient who has presented a reaction after a sting by an insect, the identity of which is either uncertain or unknown, and who shows positive in the tests for several of the venoms available. From clinical experience, we advocate a bottom-up diagnosis, starting with the information about patients and their environment and finishing with complimentary tests.

### 3.1. Visual Identification of the Insect

Among the first tools to aid diagnosis is the knowledge of the hymenoptera fauna in the patient’s region. It is accepted that *Polistes* inhabit primarily the south and *Vespula* the center and north of Europe. However, this does not rule out the presence of infrequent vespid nests in unexpected areas. Other species such as *Dolichovespula* and *Vespa*, which occupy restricted habitats, receive less attention from allergologists due to the lack of products for the treatment of patients. The recent presence in many European regions of *Vespa velutina*, with its highly invasive nature, represents a special case. So much is this so, that this insect has been reported to be involved in more than 70% of cases of anaphylaxis in one region in the north of Spain in 2018 [22].

Identification by the patient of the insect responsible for the sting or its nest may give more weight to the diagnosis and reach high levels of certainty in the case of bees, but in the case of the identification of wasps, a study has demonstrated that patients incorrectly identified insects between 27.7% and 41.9% of the time and nests between in 70.5% and 81.7% of instances [40]. In exceptional cases, the patient may capture the stinging insect, which may help (by entomologic identification) the diagnosis. A study conducted with 113 children and adults that had experienced systemic reactions following wasp stings in the Madrid area, with sensitization mainly to *Vespula* and *Polistes* venoms, concluded that the local setting of the sting (rural/urban) did not help to identify the insect responsible. However, the stings that occurred summer–autumn were significantly associated with a predominant sensitization to *Vespula*, while those occurring in the spring were associated with a predominant sensitization to *Polistes* [12]. This allowed the authors to establish a clear relationship between the seasonal presence of the insects and sIgE levels against the venom. In any case, identification of the insect that has caused an episode of anaphylaxis in a patient doubly sensitized and exposed to stings from two or more species is a poor predictor of the risk of future episodes.

### 3.2. Skin Test

Extracts of the venom from *Vespula* spp. and *Polistes* spp. are available on the European and American markets for skin tests. The extract from *Polistes dominula* is also available in Europe. The intradermal reaction with serial dilutions of venoms at subtoxic concentrations has not been shown to have sufficient sensitivity to identify the venom responsible in the case of double sensitization to vespids [15].

### 3.3. Serum Specific IgE Determinations against Whole Venom and Its Components (sIgE)

The most basic technique in allergy diagnosis that provides quantification is the determination of IgE. Currently, a quantitative interpretation of sIgE levels against the two venoms is accepted. The higher level of sIgE is attributed greater clinical relevance and the primary sensitizing factor, although no relationship between the levels of sIgE and the severity of the reaction following the sting has been established [41]. In the case of double sensitization to vespids, as in other cases, the clinically relevant cut-points remain unknown and in practice, intuitive comparisons are made between sIgE levels in the same individual to take clinical decisions based on the diagnosis. A further difficulty is that different laboratory methods may have different sensitivities and specificities [42].

The possibility of measuring sIgE against the individual components of vespid venoms lacking carbohydrates, which could give rise to confusion, has led to an increase in the number of research projects searching for key biomarkers for the differential diagnosis of *Vespula/Polistes.* The quantitative interpretation of sIgE levels against natural forms of Phospholipase A1 (nPol d 1/nVes v 1) and Antigen 5 (nPol d 5/nVes v 5) has allowed the clinical relevance of one of the two species to be assigned in 69% of cases [11]. However, this study did not define cut-off points and was performed using technology that is not commercially available. Subsequently, commercial rVes v 5 and rPol d 5 have been used to establish that a quotient greater or equal to 50% between the sIgE values against both allergens is sufficient to consider the predominant sensitization to be relevant [43]. This useful finding excludes patients sensitized to vespids with a profile of response to group 1, who were also identified in the study by Monsalve et al. [11], as currently no validated commercial product exists to test sIgE against Pol d 1 individually.

Sometimes a clinical history highly suggestive of allergy to wasp venom exists, but it cannot be confirmed by detectable levels of sIgE against the whole venom of *Vespula* or *Polistes.* In these cases, the isolated determination of sIgE against the available components 1 and 5 slightly improves diagnostic sensitivity but better results are obtained using the whole enriched venom in Antigen 5 [44]. In Europe, this diagnostic strategy is notably improved if the extract from the autochthonous *Polistes dominula* species is used rather than the American extract of *Polistes* spp. [45].

The Antigen 5 from different wasp and ant species have been studied and obtained using recombinant technology and have shown extensive cross-reactivity using sIgE inhibition and the basophil activation test (BAT) [16]. As a result, the proven usefulness for the discrimination between sensitization to bee and wasp venoms is not sufficient to determine the primary sensitizer across different vespids.

### 3.4. Serum Specific IgE Inhibition Assays (sIgE-INH)

To date, in cases of double sensitization to *Vespula y Polistes*, IgE inhibition has been found to be the most useful tool to identify the venom responsible and prescribe the most appropriate immunotherapy. A heterologous inhibition of 70–75% has been accepted as a sufficient condition to discriminate between real double sensitization and cross-reactivity and has reduced the percentage of apparent double sensitizations from 73% (*Vespula vulgaris/Polistes dominula*) or 72% (Ves v 5/Pol d 5) to 25% [15]. Furthermore, the discrimination provided by sIgE-INH has a very low correlation with that obtained by sIgE against Antigen 5 on its own, except when the sIgE value against Ves v 5 is at least double that of sIgE against Pol d 5 and vice versa [46], which is consistent with the findings of Caruso et al. [43].

### 3.5. Basophil Activation Test (BAT)

The BAT has shown its diagnostic usefulness in allergy to hymenoptera when skin tests are negative and sIgE levels negligible [47], and in those cases when neither the skin test or sIgE are conclusive to diagnose with certainty subjects doubly sensitized to bees and wasps [48]. A recent study based on the BAT with Antigen 5 from different vespids has shown different sensitivities between subjects [16], but more data are needed to establish the real value of this complex diagnostic technique in the identification of the primary sensitizing vespid in clinical practice.

### 3.6. Serum Specific IgG4 (sIgG4) as a Marker of Exposure

sIgG4 has been accepted as a marker of exposure to venom, either naturally or by stings [49], and also following immunotherapy injections [50]. This may be useful in clinical settings to circumscribe the diagnosis when the patient has experienced the sting of a non-identified wasp in an area colonized by both *Vespula* and *Polistes* and shows similar sIgE levels against the two venoms, irrespective of what the molecular profile of sensitization may be. sIgE4 levels against *Vespula* and *Polistes* may give an idea of the species that are most likely to affect subjects in their environment and serve to help make the decision on which venom immunotherapy to choose.

### 3.7. Sting Challenge Test (SCT)

In the case of food or drug allergy, the Controlled Challenge Test is accepted for the exclusion diagnosis. The SCT with live hymenoptera is the gold standard for monitoring the efficacy of VIT [50], but important problems exist for its diagnostic use. In the case of double sensitization to vespids, the cross-reactivity between the venoms of both species may yield positive responses to either of them, submitting the patient to an unacceptable risk. Furthermore, we know that a single isolated sting does not predict the persistence of negative results [51], and currently the use of this technique for merely diagnostic purposes is not recommended [13].

## 4. Management of Vespid Venom Immunotherapy (vvit) in Cases of Double Sensitization

VVIT accounts for 70% of all that is published on immunotherapy with hymenoptera venom [41] and its use is a challenge in the case of patients with a double sensitization as therapeutic success requires that the choice of venom is appropriately aligned with diagnostic accuracy [50]. When the uncertainty regarding the insect responsible is high, wide coverage therapy may be necessary.

The vespid extracts that are used in Europe for subcutaneous immunotherapy are available in non-purified aqueous preparations for *Vespula* spp., *Polistes* spp. and *Polistes dominula* and aqueous purified and depot preparations, adsorbed with aluminum hydroxide only for *Vespula* spp. [50]. In Italy, there exists a product for *Vespa crabro* and as far as we know there is no commercially authorized product for treatments with *Dolichovespula*. It is common in the extracts of *Vespula* to use mixtures from various species (*V vulgaris*, *V flavopilosa*, *V germánica*, *V maculifrons*, *V pennsilvania y V squamosa*) with a high degree of cross-reactivity, with the aim of covering a wide range of sensitivities depending on the patient’s geographic region [6]. In the case of *Polistes* extracts, a mixture of American Polistes (*P annularis*, *P fuscatus*, *P metricus y P exclamans*) was the only treatment available up to 1996 when the production of de *Polistes dominula* venom began, which is currently available for diagnosis and treatment in European countries. This specific *Polistes dominula* extract has shown significant differences with the extract from American species with regard to its efficacy [6,41].

The standardization of vespid extracts is based on a differential allergen, not present in ant venom (Phospholipase A1) and also cross-reactivity (Hyaluronidase) as well as overall protein content. No control for Antigen 5 exists as its function is unknown and commercial vespid extracts with specified individual allergens and controlled lot by lot are not available [52].

As has been stated, cross-reactivity is the cause of apparent multiple sensitization. This has been demonstrated for the genera of *Vespula*, *Dolichovespula* and *Vespa* of the *Vespinae* subfamily [27] and particularly between *Vespula germanica* and *Vespa crabro* [53]. It is a fact that patients principally sensitized to *Vespula* may develop severe anaphylaxis following a sting from *Vespa crabro* and vice versa and furthermore, that patients allergic to *Vespa* may be adequately treated with extracts from *Vespula* [15,53]. However, a recent article has suggested that, in patients with a reaction caused by a sting from *Vespa crabro*, immunotherapy with venom from *Vespa crabro* may have a higher safety profile [54]. Moreover, inhibition studies conducted on a small number of patients with anaphylaxis due to *Vespa velutina* seem to show that, at least for the moment and in the population studied, *Vespula* behaves as the primary sensitizer [33].

Thus, the use of a *Vespula* vaccine would seem advisable or recommended in these patients as, indeed, has already been demonstrated in a group of patients in whom the administration of *Vespula* venom induced immunologic changes (reductions in sIgE and increases in sIgG4) after one year of treatment and protection against subsequent spontaneous stings from the hymenoptera [18]. In a small group of patients allergic to *Vespa orientalis* with unequivocal identification of the insect, a positive skin test to the extract of the venom from the venom sac and to *Vespula* venom in the absence of a commercial *Vespa orientalis* extract, treatment with extract from *Vespula* spp. has proved to be effective [55].

In European patients, there may be a difference in the protective effect of *Polistes* spp. extract with regard to *Polistes dominula* due to the existence of low cross-reactivity between the allergens of the European (*P dominula* and *P gallicus)* and the American species *(P annularis*, *P apachus*, *P exclamans*, *P fuscatus y P metricus)* [6,43,55]. However, an Italian study has confirmed that although the cross-reactivity measured in vitro is incomplete, it did not detect differences in the protection between the immunotherapy with the mixture of American species and the immunotherapy with *Polistes dominula* following spontaneous stings [56]. Due to limitations in design, specific efficacy studies are needed in the European population to confirm this finding.

The maintenance dose recommended for VIT es 100 µg of protein, the equivalent to the dry weight of approximately five wasp stings [50]. The use of a dose of 200 µg of venom from *Vespula* spp. has been explored in the case of therapeutic failure and a more intense immunologic response was observed with regard to the skin test and falls in sIgE levels as compared to patients treated with 100 µg [57]. Increasing the dose above 100 µg could be an appropriate approximation to the treatment with *Vespula* venom of patients with anaphylaxis following *Vespa* stings due to its greater size and subsequent capacity to inject a larger volume of venom with its sting.

In regions where *Vespula* and *Polistes* coexist, double sensitization is not always the consequence of cross-reactivity. When double sensitization is certain or if a high degree of diagnostic uncertainty exists, the need for double vaccination must inevitably be considered [11,57] The lack of evidence concerning possible enzymatic degradation means that it is not advisable to mix the venoms from *Vespula* and *Polistes* in the same vial [52]. Additionally, the separate administration of the two extracts during the maintenance phase increases costs and hinders patient accessibility. However, a therapeutic strategy that has been successfully used for years by Spanish allergologists is the alternate month administration of pure extracts from *Vespula* and *Polistes* at full doses (Figure 3). This has been shown to maintain the same safety and efficacy, as confirmed by intra-hospital re-sting tests, as conventional monthly treatment given as monotherapy, irrespective of the insect responsible for the reaction [58].

Patients with reactions to wasp stings and double sensitization to vespid venom are managed in clinical practice with clearly limited resources. There is not perfect diagnostic tool; doctors have the tools described to identify the culprit insect, but when all else fails, the latest proposal is double vaccination. Prospective and multi-center studies are needed to increase the evidence base and to provide firmly based recommendations.

## Figures and Tables

**Figure 1 toxins-14-00126-f001:**
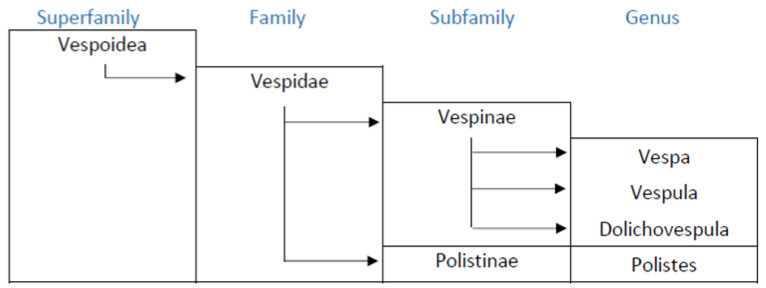
Taxonomic relationship among main allergenic wasps.

**Figure 2 toxins-14-00126-f002:**
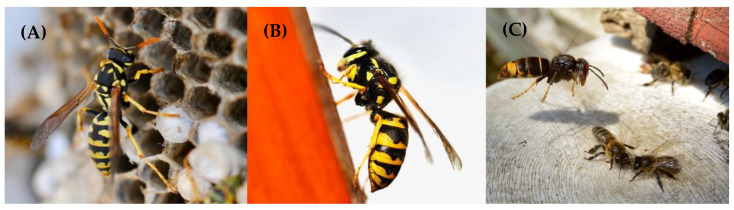
(**A**): *Polistes gallicus* in its nest. (**B**): *Vespula germanica* sipping nectar. (**C**): *Vespa Velutina* hunting bees in a hive.

**Figure 3 toxins-14-00126-f003:**
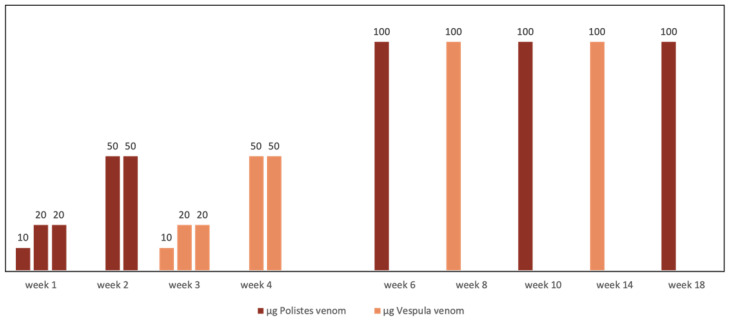
Scheme of alternate full dose administration of *Vespula* spp. and *Polistes dominula* venom used for doubly sensitized patients.

**Table 1 toxins-14-00126-t001:** Characteristics of the main known allergens from Vespula germanica and Polistes dominula.

Allergens	% Sensitization [23]	kDa	Isoforms	Glycosylation	Sequence Homology	Eukaryotic Expression
Ves v 1Phospholipase A1	33.3–54	35	1	no	Ves v 1 y Ves f 1, Ves g 1, Ves m 1, Ves s 1, Pol a 1, Pol d 1, Dol m 1, Ves p 1, Vesp c 1, Poly p 1 y Sol i 1	yes
Ves v 2Hyaluronidase	5–25	44	2	yes	Api c 2, Api m 2, Dol m 2, Pol a 2 y Ves g 2	yes
Ves v 3Dipeptidylpeptidase IV	50–62.8	100	1	yes	Pol d 3, Api m 5	yes
Ves v 5Antigen 5	84.5–100	23	1	no	Dol a 5, Dol m 5, Pac c 3, Pol a 5, Pol d 5, Pol e 5, Pol f 5, Pol g 5, Poly s 5, Sol i 3, Sol r 3, Ves f 5, Ves g 5, Ves m 5, Ves p 5, Ves s 5, Ves vi 5, Vesp c 5, Vesp m 5.	yes
Ves v 6Vitellogenin	39	200	1	yes	Api m 12	yes
Pol d 1Phospholipase A1	87	33	4	no	Dol a 5, Dol m 5, Pac c 3, Pol a 5, Pol d 5, Pol e 5, Pol f 5, Pol g 5, Poly s 5, Sol i 3, Sol r 3, Ves f 5, Ves g 5, Ves m 5, Ves p 5, Ves s 5, Ves vi 5, Vesp c 5, Vesp m 5.	no
Pol d 2Hyaluronidase	?	50	1	yes		yes
Pol d 3Dipeptidylpeptidase IV	66.7	100	1	yes		yes
Pol d 4 Serinprotease	?	30	1	yes	Api m 7, Bom ig 4, Bom p 4, Bom t 4	no
Pol d 5Antigen 5	69–72	23	1	no	Dol a 5, Pac c 3, Poly s 5, Sol i 3, Ves v 5, Vesp c 5	no

**Table 2 toxins-14-00126-t002:** Sequence identity of the main allergenic proteins of Vespula germanica and Polistes dominula [24].

	Ves G 1	Ves G 2	Ves G 3	Ves G 4 *	Ves G 5
Pol d 1	54%				
Pol d 2		74.1%			
Pol d 3			73.7%		
Pol d 4				51.9%	
Pol d 5					59.6%

(*) Hypothetical protein.

**Table 3 toxins-14-00126-t003:** Degrees of sequence identity (%) for vespid Phospholipases 1.

**Ves v 1**	**Ves g 1**	**Ves m 1**	**Ves s 1**	**Pol a 1**	**Pol d 1**	**Dol m 1**	**Vesp c 1**	**Vesp v 1**	**Poly p 1**
	94.7	95.7	71.1	53.8	52.8	67.2	72.1	62.5	61.7

## Data Availability

No new data were created or analyzed in this study. Data sharing is not applicable to this article.

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
