# Peer review of "Management of Double Sensitization to Vespids in Europe"

_toxins, 2022, doi:10.3390/toxins14020126_

Round 1

Reviewer 1 Report

This review article is focused on the challenges and manners of the management of double sensitization to vespids (in particular to Vespula and Polistes). The Authors thoroughly described the known allergens present in vespids. They also addressed allergy diagnosis to venoms of different species and treatment by vespid venom immunotherapy. The article is well written and provides structured information on double sensitization to vespids.

I have a few comments:

References should support the information presented on lines 25-36.

Lines 33-36: It’s worth adding a simple cladogram showing the relationships between selected genera and species belonging to the order Hymenoptera. This will make it easier for the reader to understand the introduction.

Line 79: To enrich the manuscript, sequence alignment of the chosen allergens should be presented. Maybe also the visualization of the structures of the selected allergens? Is it possible to visualize the differences between allergens found in the species of interest?

Lines 121-133: I miss the % similarity of hyaluronidase forms between the species in the text (although presented in Table 3).

The description of Table 4 should be added (with an explanation).

Paragraph 4: The authors could present data on the efficacy of immunotherapy according to species and the extract used (if available).

Author Response

References should support the information presented on lines 25-36. Included

Lines 33-36: It’s worth adding a simple cladogram showing the relationships between selected genera and species belonging to the order Hymenoptera. This will make it easier for the reader to understand the introduction. Included as Figure 1

Line 79: To enrich the manuscript, sequence alignment of the chosen allergens should be presented. Maybe also the visualization of the structures of the selected allergens? Is it possible to visualize the differences between allergens found in the species of interest? The focus of this manuscript is clinic. We apply for a monographic issue in which, allergenic proteins will be widely considered.

Lines 121-133: I miss the % similarity of hyaluronidase forms between the species in the text (although presented in Table 3). Data included (now, Line 196)

The description of Table 4 should be added (with an explanation). Table IV has been eliminated

Paragraph 4: The authors could present data on the efficacy of immunotherapy according to species and the extract used (if available). There are not comparable data.

Reviewer 2 Report

Although we can appreciate the Authors effort for the work done, the manuscript is not well presented, thus this does not allow its correct evaluation. The manuscript lacks the perspectives for overcoming the state of the art. Some points are unclear. Just a few examples:

In the table 1 there is a percentage of sensitization, what is the considered population?

The allergen name or the name of the corresponding protein are used interchangeably, choosing a criterion could help the reader in understanding.

Line 103: Their biological action is important for the allergenic characterization of venoms. What does it means actually?

Line 114: …phospholipase A1…from which hymenoptera?

Author Response

In the table 1 there is a percentage of sensitization, what is the considered population? These data are referred to reference 23

The allergen name or the name of the corresponding protein are used interchangeably, choosing a criterion could help the reader in understanding. Table 1 has been modified in its first column, to a better reading

Line 103: Their biological action is important for the allergenic characterization of venoms. What does it means actually? Allergenic vaccines are made from the source of natural venoms. Identification of individual proteins based on the biological activity is a key point in order to standardize the content batch to batch.  

Line 114: …phospholipase A1…from which hymenoptera? Vespula and Polistes (now Lines 181-182)

Reviewer 3 Report

What kind of study is this? A narrative review? 

What is the objective? 

Which methods were used?

Conclusions? 

Tables need proper titles.

What exactly is table 4 about? Who are those patients?

Author Response

What kind of study is this? A narrative review? The term “narrative review” has been included in the paragraph of Lines 103-105

What is the objective? This paragraph (lines 103-105) has been included to clarify the objective.

Which methods were used? Methods are implicit in the term “narrative review”, but we can develop this issue if required.

Conclusions? Last paragraph (Lines 438-442) has been extended, to clarify conclusion

Tables need proper titles. Included

What exactly is table 4 about? Who are those patients? Table 4 has been eliminated

Round 2

Reviewer 2 Report

I really appreciated the efforts of the authors in addressing the criticisms raised.

 I suggesting just a couple of revisions:

  1. I suggest to revise this sentence lines 192-194: Their biological action is important for the allergenic characterization of venoms. What does it means actually?
  2. I suggest a careful revision of English language (line 84 correct“frecuent”)

Author Response

  1. I suggest to revise this sentence lines 192-194: Their biological action is important for the allergenic characterization of venoms. What does it means actually. An explicative paragraph has been included
  2. I suggest a careful revision of English language (line 84 correct“frecuent”) Made

Reviewer 3 Report

Is there a table 2?  I found a table 1 and a table 3.

There are three figures, the third is wrongly identified.

Correction line 84 is needed.

I don't agree there is no need to describe methods once you mentioned a narrative review. You should mention briefly how the search was done: words, data, years included, language...for readers to have an idea of how deep your review was.

Author Response

Is there a table 2?  I found a table 1 and a table 3. Corrected

There are three figures, the third is wrongly identified. Corrected

Correction line 84 is needed. Corrected

I don't agree there is no need to describe methods once you mentioned a narrative review. You should mention briefly how the search was done: words, data, years included, language...for readers to have an idea of how deep your review was.  An explicative paragraph has been included (Lines 106-112 last version)